## Perspective

saltmarsh restoration; sedimentation fields; human influence

**Corresponding author:**
Jonathan Dale;
Email: j.j.dale@reading.ac.uk

# Sedimentation fields as a method of saltmarsh restoration: Continuity of human influence on natural processes

Jonathan Dale[1] 🔘 and Michelle Farrell[2]

[1]University of Reading, UK and [2]Coventry University, UK

## Abstract

Saltmarsh habitat provides important ecosystem services, such as water quality regulation, carbon sequestration and flood defence, but is experiencing losses globally. Historically, this has been caused by land claim, and more recently by rising sea levels. Several methods have been implemented to compensate for saltmarsh habitat loss, including realigning defences, transplanting vegetation, and building structures such as sedimentation fields to enclose areas of mudflat and encourage sediment deposition. It has been suggested that sedimentation fields may offer saltmarsh restoration without the limitations identified in other restoration approaches, such as poor drainage and anoxia caused by changes to the sediment structure due to prior human activity. In this article, we argue that restoration through sedimentation fields should be viewed as a continuation of human activity influencing natural processes, rather than a method that overcomes the influence of prior human activity on saltmarsh ecosystem functioning. This opinion is evidenced by a critical review of the (pre-)historic human activity and saltmarsh restoration attempts at Rumney Great Wharf, Severn Estuary, Wales, where sedimentation fields were constructed between 1989 and 2005 and extended in 2024. We then evaluate the research requirements that need to be addressed to ensure the successful implementation of future schemes, including further understanding of the interactions between physical and biological processes, to enhance ecosystem functioning in sites restored using sedimentation fields.

## Impact statement

Sedimentation fields are used to restore or create saltmarsh habitat by enclosing areas of mudflat, often using brushwood fencing, with the intention of dampening waves and reducing current velocities to encourage sediment deposition and saltmarsh colonisation. This approach to saltmarsh restoration can be implemented where it is not possible to use other methods, such as managed realignment – the breaching, lowering or removal of flood defences to allow tidal inundation of the coastal hinterland. For example, sedimentation fields do not require the purchase of terrestrial (typically agricultural) land and conversion through engineering works to make it suitable to support intertidal habitat, which can be costly and socially sensitive. It has also been proposed that sedimentation fields might provide saltmarsh restoration without the limitations associated with prior human activity found in other restoration approaches. However, in this article, we argue that sedimentation fields should be seen as an extension of human influence on natural processes, rather than reversing the impact of human activity on saltmarsh ecosystem functioning. Evolution of sites restored using sedimentation fields can still be affected by site history, including both natural and anthropogenic influences. Depending on the targeted outcome of the scheme, it is important that sedimentation field construction results in the formation of a saltmarsh that provides the required ecosystem functioning and service provision. This could, for example, include increased flood protection, carbon storage or the creation of a diverse saltmarsh habitat. Further research is needed into the physical processes, ecological development and biogeomorphic evolution of sedimentation fields to inform the design and implementation of future schemes.

## Compensating for saltmarsh habitat loss

Saltmarsh habitat occupies ~5.1 Mha of the Earth's surface (Pendleton et al., 2012) and provides important ecosystem services, such as carbon sequestration and flood defence through wave attenuation (e.g., Costanza et al., 1997). Globally, 46.6% of saltmarsh has been lost or degraded (Brook et al., 2025) due to historic land claim (e.g., Allen and Fulford, 1986) and modern sea level rise resulting in erosion and coastal squeeze (e.g., Doody, 2004). However, during the mid-twentieth century, the approach to saltmarsh management shifted from reclamation to

protection, and then in the twenty-first century to restoring and increasing saltmarsh extent (Ladd, 2021).

Several approaches have been adopted to restore and (re)create saltmarsh (Pontee et al., 2021). These include allowing tidal inundation of terrestrial, often (re)claimed, land through either regulated tidal exchange (e.g., Masselink et al., 2017) or breaching and lowering of defences via managed realignment (e.g., Schuerch et al., 2022). However, purchasing land and landscaping sites so they can support the targeted habitat types can be costly, socially unacceptable and culturally sensitive (e.g., Ledoux et al., 2005; Yamashita et al., 2019). Alternatively, in situ restoration methods that do not require the purchase and conversion of terrestrial land, such as transplanting vegetation (Taylor et al., 2019), are available. Sediment accumulation can also be encouraged using revetments and other structures, such as coir rolls, sandbags, silt fences and sedimentation fields to trap sediment and enhance sedimentation rates (e.g., Henry et al., 1999; Siemes et al., 2020; Vona et al., 2020; Reeder et al., 2021; Cox et al., 2022; Gonçalves et al., 2025), or can take place artificially through the deposition of dredge material (e.g., Baptist et al., 2019).

These methods have proven to be relatively successful, with 1,279.84 km$^2$ of saltmarsh gained globally through restoration between 2000 and 2019, although this is not yet equivalent to the 2,733.33 km$^2$ lost (Campbell et al., 2022). Of the various saltmarsh restoration methods, the most common, and therefore the most researched, has been managed realignment. Managed realignment sites have been shown to offer benefits such as carbon storage and habitat for fish (e.g., Burgess et al., 2020; Mossman et al., 2022; McMahon et al., 2023). However, they have a lower diversity and abundance of key plant species, and increased amounts of unvegetated ground, in comparison to pre-existing reference marshes (e.g., Mossman et al., 2012). It has been argued that these differences result from the sites' former land use, which is typically agricultural, causing irreversible alterations to the (re)claimed sediment (e.g., Spencer and Harvey, 2012). This leads to reduced hydrological connectivity following site breaching (Tempest et al., 2015), reduced aeration of the sediment and anoxia (e.g., Spencer et al., 2017; Van Putte et al., 2025). Managed realignment sites also have reduced topographic complexity and fewer elevational niches (e.g., Lawrence et al., 2018), resulting in lower plant diversity (e.g., Morzaria-Luna et al., 2004).

There is a need for increased large-scale saltmarsh restoration and creation efforts that mitigate the issues related to prior human activity and site history identified in managed realignment. One potential solution is the construction of sedimentation fields, which have been used historically in the Wadden Sea (Dijkema et al., 2001; Esselink et al., 2011), more recently in locations such as Venice Lagoon, the Ems-Dollard estuary and the North Sea (Scarton et al., 2000; Reeder et al., 2021; Siegersma et al., 2023), and for mangrove restoration (Winterwerp et al., 2020). This involves (partially) enclosing an area of intertidal mudflat with the intention of trapping sediment by reducing tidal currents and wave energy, increasing sediment deposition and facilitating saltmarsh creation (Pontee et al., 2021). Consequently, costly site purchases and landscaping works are not required, and the evolution of the marsh should not be affected by site history, the trade-off being a loss of mudflat habitat. In this article, we argue that this perception is a false distinction, and the construction of sedimentation fields is, in fact, the continuation of human activity influencing natural processes.

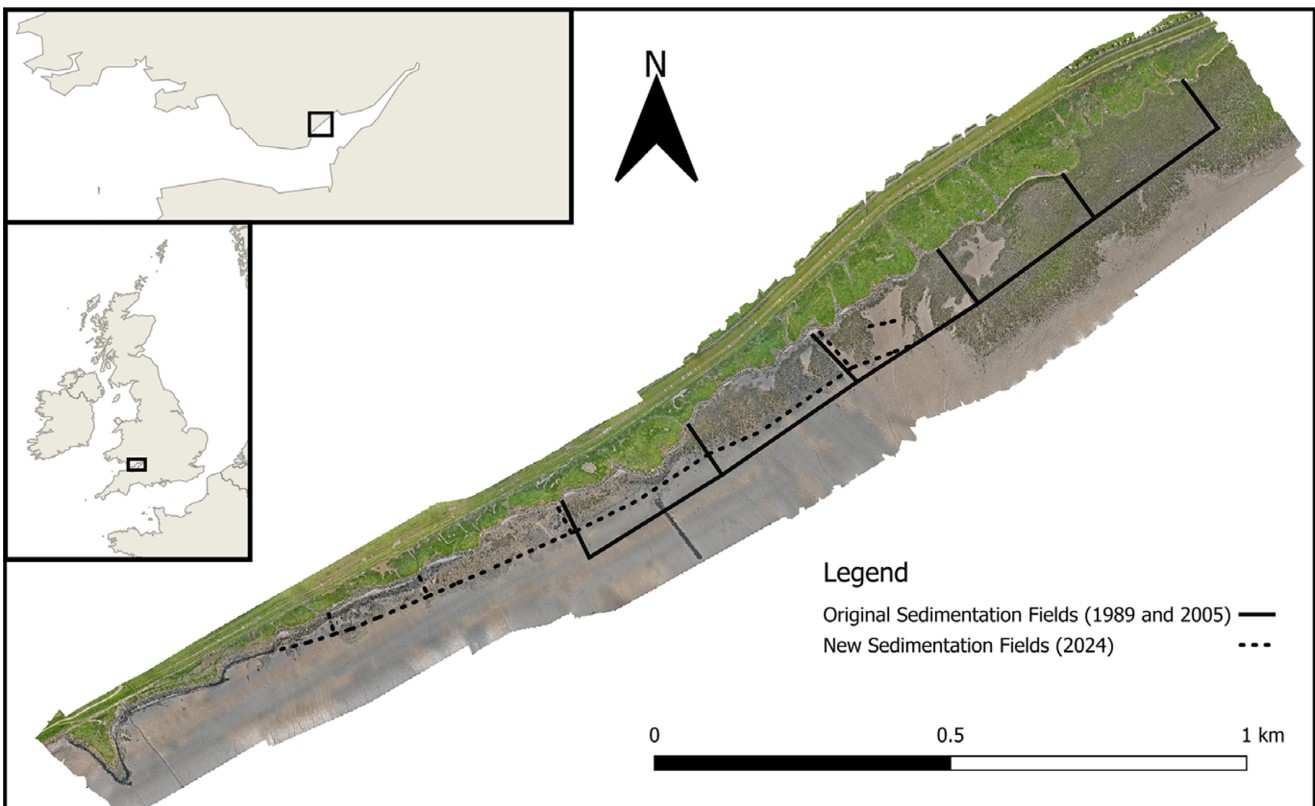

**Figure 1.** Aerial imagery of Rumney Great Wharf sedimentation fields, collected by the authors using an uncrewed aerial system, showing the locations of the originally installed and new sedimentation fields. The most easterly of the original sedimentation fields was installed in 1989, with the remaining four installed in 2005. The new sedimentation fields were installed in 2024. The regional (upper insert) and national (lower insert) settings are also indicated.

We argue this using the case of Rumney Great Wharf, Severn Estuary, Wales (Figure 1), where sedimentation fields were constructed between 1989 and 2005, and extended in 2024. Rumney Great Wharf allows for temporal comparisons of recent restoration efforts in the context of current shoreline management policies and approaches. We then evaluate the future evidence needs that should be considered to ensure the effective implementation and success of future restoration and creation attempts through sedimentation field construction.

## Human activity and landscape evolution at Rumney Great Wharf

### Holocene evolution

Rumney Great Wharf's modern-day landforms and landscape features result from the intrinsic and long-term relationship between human activity and landscape evolution (Figure 2). Bell et al. (2000) described the general Holocene sedimentary sequence of the Severn Estuary, which comprises a series of layers of estuarine sediments and peats. The uppermost of these layers is exposed at low tide, and more recent deposits have often been eroded. The oldest layer is known as the Wentlooge Formation (Allen and Rae, 1987) and comprises three subdivisions: the lower, middle and upper Wentlooge formations. The lower Wentlooge Formation consists of estuarine clays and silts and formed during a period of rapid sea level rise during the early Holocene (Bell and Neumann, 1997). Analysis of pollen and foraminifera contained within these sediments indicates that the site supported mudflats with some localised areas of saltmarsh (Green, 1989).

The middle Wentlooge Formation was deposited following the deceleration in the rate of sea level rise around 6,000 BP. The formation consists of peat intercalated with estuarine clays and silts, and continued to form until 2,500 BP (Allen, 1997) when the upper Wentlooge Formation was laid down following a widespread marine transgression. The upper Wentlooge Formation consists of marine silts and clays and can be up to 4.5 m in depth (Bell et al., 2000). There have been subsequent periods of erosion and deposition, with three intertidal formations being deposited since the medieval period. The earliest of these is the Rumney Formation, dating to the thirteenth to fourteenth centuries (Allen, 1987). The second deposit, the Awre Formation, was laid down in the late nineteenth century, with the Northwick Formation being deposited during the twentieth century (e.g., Allen, 1997).

### Prehistoric and historic (4150–800 BP) human activity

The earliest evidence of human activity at Rumney Great Wharf includes the construction of fish traps and livestock pens (Nayling, 1991) during the Bronze Age (4150 to 2750 BP). However, the first evidence of large-scale landscape change due to human activity occurs alongside evidence of Iron Age and Roman activity,

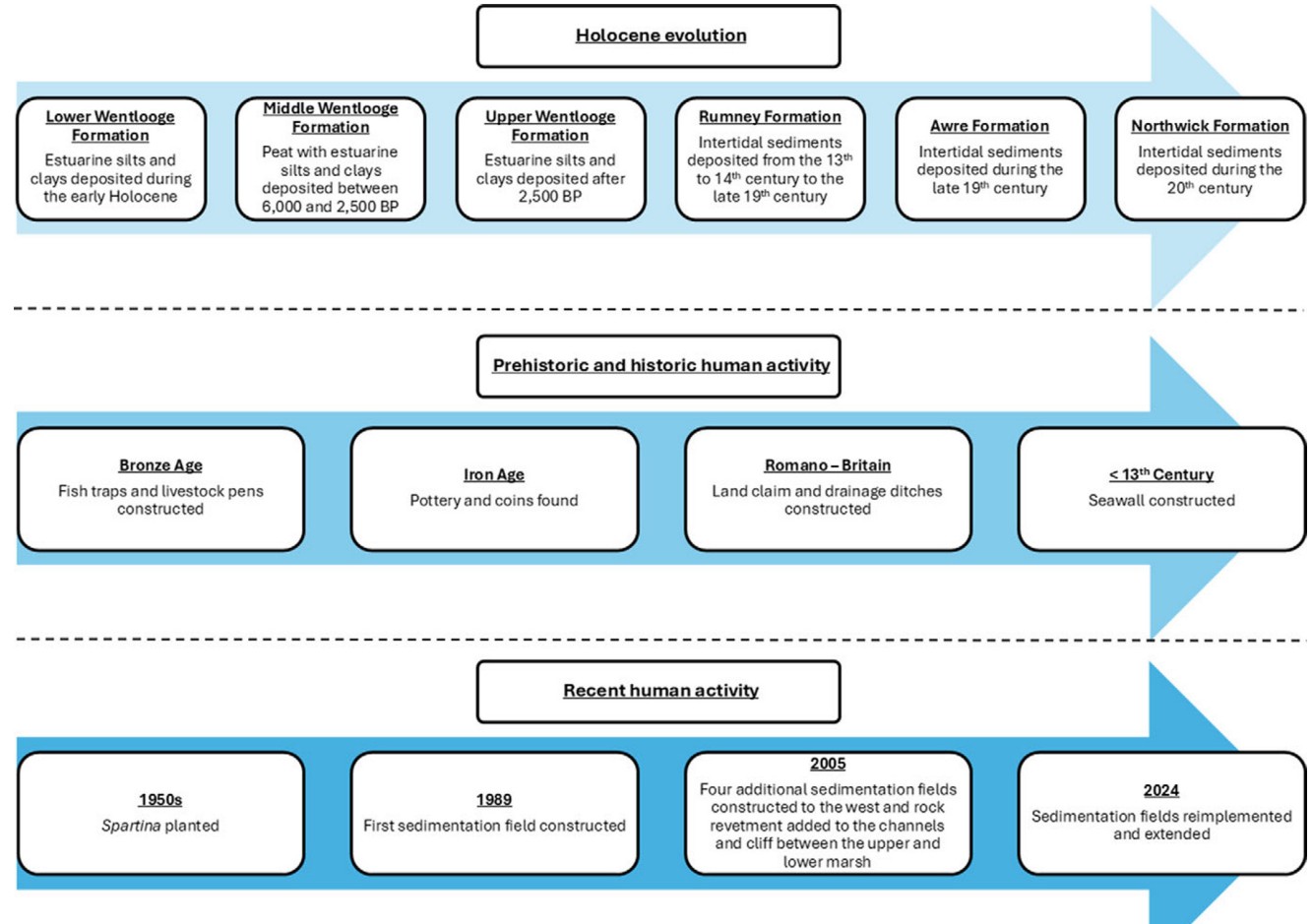

**Figure 2.** Schematic representation of the landform evolution (top), prehistoric and historic human activity (middle) and recent human activity (bottom) at Rumney Great Wharf.

including pottery and coins (e.g., Fulford et al., 1994). Specifically, Allen and Fulford (1986) propose that a series of drainage ditches containing Romano-British pottery is indicative of the first planned reclamation attempts at the site.

Subsequent phases of reclamation and landscape change occurred, including the construction of an earth embankment seawall before the thirteenth century AD, which prevented deposition of the more recent Rumney, Awre and Northwick formations on the landwards side (e.g., Fulford et al., 1994). The resulting modern-day intertidal landscape (Figure 3) consists of areas of saltmarsh formed on the Rumney, Awre and Northwick formations, with the peats of the middle Wentlooge Formation exposed to the west of the saltmarsh due to erosion (Figure 3b). In isolated areas, saltmarsh has colonised the exposed peat. Behind the modern seawall is the Gwent Levels, a reen and ditch habitat formed through intertidal land claim. This area is a Site of Special Scientific Interest and supports rare and endangered insect species such as *Odontomyia ornata* and *Hydaticus transversalis*.

## Recent (mid-twentieth century to present) human activity

Since the mid-twentieth Century, there has been a trend of erosion and saltmarsh loss at Rumney Great Wharf. It is estimated that the foreshore has lowered at a rate of 0.01–0.04 mm per year since 1965 (Armstrong et al., 2021). There have been several attempts to prevent and reverse these losses. In the 1950s, a *Spartina* planting campaign took place to try and stabilise the saltmarsh (ABPmer, 2009) – the success of which remains unknown, as no subsequent monitoring work was undertaken. Between 1989 and 2005, brushwood sedimentation fields were constructed (Figure 1). Initially, in 1989, a single sedimentation field was constructed in an area of mudflat to the west of, and fronted by, a pre-existing saltmarsh. The fencing was subsequently extended westwards in 2005, creating five sedimentation fields. Rock revetment was also added in 2005 to stabilise the channels and cliff between the upper and lower marsh, which in some places is 3 m high. Since construction, saltmarsh has developed in the eastern part of the sedimentation fields, where elevation has increased by over 0.5 m (Armstrong et al., 2021).

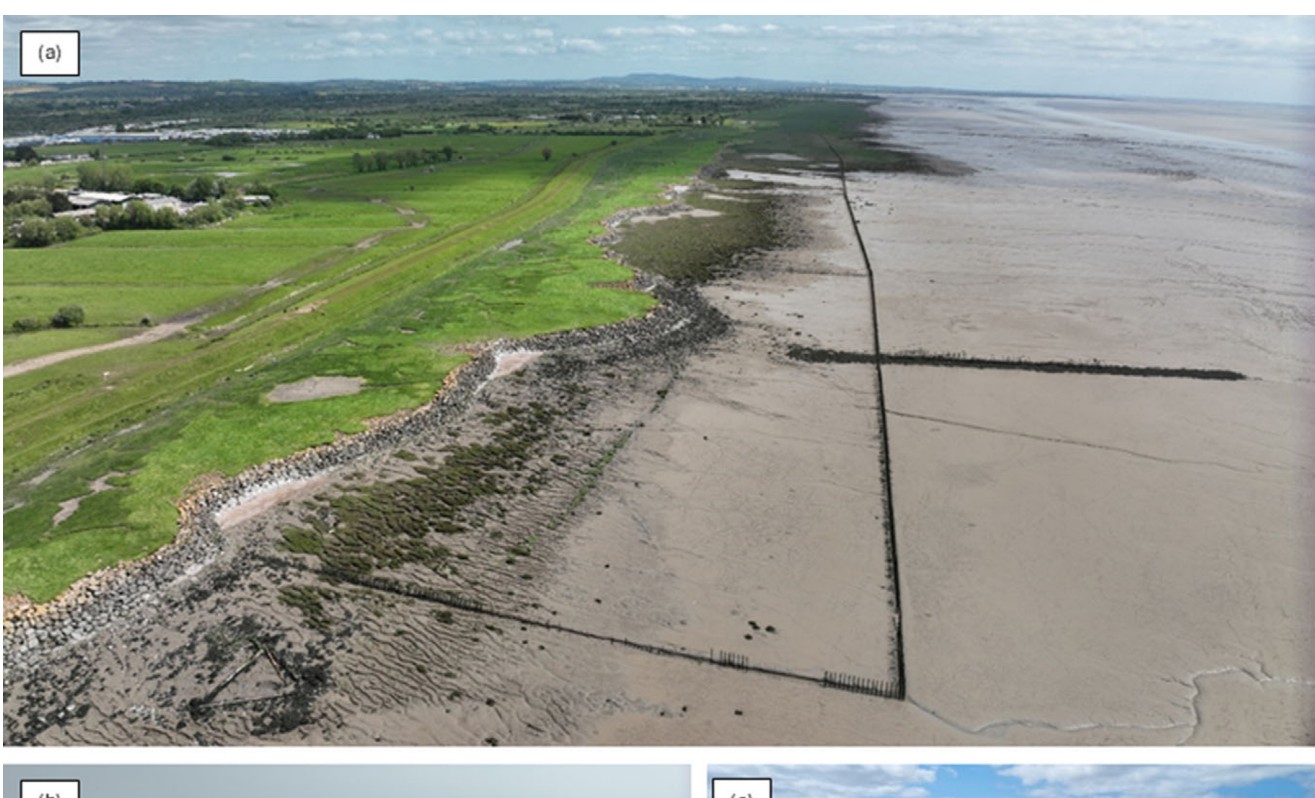

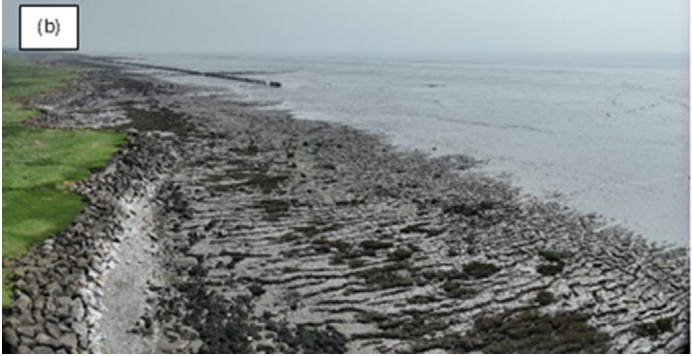

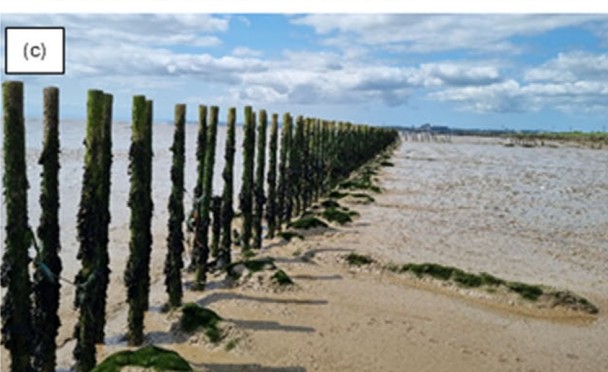

**Figure 3.** Imagery taken by the authors in May 2023 of (a) the western end (looking eastwards) of the Rumney Great Wharf sedimentation fields, including the remaining posts from the original brushwood fencing and the seawall with the Gwent Levels behind to the left of the image, (b) the exposed peats of the middle Wentlooge Formation to the west of the sedimentation fields and (c) the in situ posts remaining from the original brushwood fencing construction.

However, the fencing was not maintained or routinely monitored since 2010 and has subsequently been eroded, leaving just the support posts in situ (Figure 3c).

Due to concerns over the continued flood defence capabilities of the seawall, in addition to habitat loss, Natural Resources Wales reviewed the management of the site and the possibilities for saltmarsh restoration (Armstrong et al., 2021). Re-implementing and extending new sedimentation fields was identified as the most suitable option, with the new fencing constructed between July and September 2024.

## Human activity continues to influence natural processes

The requirement to defend former saltmarsh, which has been drained for terrestrial use, through further saltmarsh creation within mudflat habitat presents an intriguing paradox. This is enhanced by the ecological significance and statutory designations of the drained saltmarsh behind the seawall, driving the need to reinforce flood defences and maintain a "Hold the Line" approach to coastal management. Restoring and compensating for saltmarsh loss is required legally, and to prevent and mitigate the impacts of climate change through carbon sequestration and flood defence (e.g., Kiesel et al., 2022; Mossman et al., 2022; McMahon et al., 2023). Sedimentation fields provide an opportunity to do this in locations where other methods, such as managed realignment, are not possible. This is usually due to a need to maintain defences to protect designated habitats or infrastructure, cost or negative public opinion (e.g., Ledoux et al., 2005; Esteves, 2014) and may help to reduce the net loss of saltmarsh (e.g., Campbell et al., 2022). In addition, sedimentation fields do not have the same limitations on saltmarsh ecosystem functioning imposed by former agricultural land-use at managed realignment sites.

Past human activity is still, however, likely to have influenced site formation processes, and hence the modern-day landscape, and should be accounted for when setting restoration targets, such as the intended saltmarsh extent or habitat type. For example, at Rumney Great Wharf, if restoration were to target similar habitats to those present before any human activity took place at the site during the Bronze Age, then it would likely be targeting a similar environment to either the lower or middle Wentlooge formations. This would result in the restoration of either the mudflat with some areas of saltmarsh (lower Wentlooge), and an environment not too dissimilar to the modern-day landscape or peats with some intertidal areas (middle Wentlooge). If the habitats before Romano-British reclamation attempts were targeted, then the marine silts and clays of the upper Wentlooge Formation could be recreated. Under both scenarios, the subsequent site evolution alongside the influence of later human activity, including the deposition and (in places) erosion of the Rumney, Awre and Northwick formations on the seaward side of the seawall (Allen, 1987; Allen, 1997; Bell et al., 2000), would not be accounted for. Restoration to a defined (pre)historic baseline is likely to be impossible given current and future projected sea level rise, and restoration targets should rather consider the most appropriate habitat types needed to meet the objectives of the scheme.

## Future sedimentation fields: Evidence requirements

As demonstrated for Rumney Great Wharf, restoration using sedimentation fields is not necessarily without the influence of previous human activity and can arguably be viewed as the continuation of human manipulation, albeit with a step-change towards encouraging saltmarsh growth. Therefore, it should be recognised that the use of sedimentation fields represents the management of hydrological functioning and sedimentation processes to artificially encourage saltmarsh creation rather than restoring natural functioning or processes and allowing the saltmarsh to develop. It also remains unclear whether saltmarsh created using this method can be self-sustaining, or whether continued maintenance is required. However, it remains important that the required and/or targeted ecosystem functioning and services, such as wave attenuation, carbon storage, habitat extent and species diversity, are provided.

To date, the limited number of studies assessing sedimentation fields have focused on predicting site evolution using numerical models (e.g., Dao et al., 2018; Siemes et al., 2020), rather than empirical observations. Those studies that have presented field-based evaluations of sedimentation fields have tended to focus on mangroves (e.g., Winterwerp et al., 2020), relatively small-scale structures built within or as an extension of pre-existing marsh (e.g., Scarton et al., 2000; Gonçalves et al. 2025) or the establishment of pioneer vegetation (Siegersma et al., 2023). Our evaluation of Rumney Great Wharf has demonstrated that using sedimentation fields for saltmarsh restoration may not result in the development of the equivalent habitat the site used to support or would support without human influence. Consequently, it is essential that the baseline habitat creation requirements are established before site implementations. While this is also true of other methods of restoration, including managed realignment, many managed realignment sites are designed and landscaped to deliver a particular habitat type or function (e.g., Pearce et al., 2011; Burgess et al., 2014). Provision of a particular target, service or deliverable is often used as the mechanism to justify and fund saltmarsh restoration schemes, and this approach is likely to continue given the move towards funding restoration through carbon finance mechanisms (Mason et al., 2022; Burden et al., 2023).

Regardless of the specific drivers, restoration is likely to have multiple benefits, and it is important that sites support a functioning ecosystem (Billah et al., 2022). Consequently, it is crucial that sedimentation fields are monitored to evaluate ecosystem functioning, including empirical assessments of the biogeomorphic processes. Specifically, sedimentation fields might provide benefits, such as the trapping of suspended contaminants, as demonstrated for silt fences used to retain toxic particles from dredge spoil (Henry et al., 1999). The formation of morphological features, the supply of sediment, ecological succession and the role of stochastic events such as storms (Scarton et al., 2000) also require further evaluation to fully understand site development. For example, studies into the relationship between sediment accretion and plant colonisation (Brückner et al., 2020) will inform assessments of saltmarsh establishment and development. In addition, topographic complexity and the formation of a range of elevational niches are known to play an important role in increasing plant diversity (e.g., Morzaria-Luna et al., 2004; Kim et al., 2013). Understanding if (and when) topographic variability develops as sediment accretes will help inform evaluations of saltmarsh development and functioning, including the level of flood defence provided by the scheme (e.g., Rupprecht et al., 2017). These data can then be used to inform the management and maintenance of sedimentation fields following construction, informing the design of future restoration schemes to maximise ecosystem service delivery.

In addition to assessments of the site itself, the benefits of sedimentation field construction on pre-existing coastal flood defences, management approaches and the wider ecosystem should

also be considered. Sedimentation fields may provide opportunities to restore saltmarsh in locations where coastal flood defences cannot be breached and flood defence standards need maintaining (Pontee et al., 2021). This includes locations where there is a need to protect infrastructure or areas of rare and important habitat, but without necessarily upgrading or investing in seawalls and hard defences. Consequently, the use of sedimentation fields for both flood defence and saltmarsh restoration is likely to increase. However, the required size and rate of marsh development needed to deliver the necessary attenuation in wave energy requires further investigation.

Alongside uncertainty relating to the level of flood defence provided, it is unknown if or when restored marshes in sedimentation fields become self-sustaining or whether there needs to be a continued commitment to maintaining the fencing and structures. Without continued maintenance, there is potential for the restored marsh to start eroding and be lost unless a certain vegetation cover threshold is reached (e.g., D'Alpaos, 2011). It also remains unclear who is responsible for funding this maintenance, and what the public perception will be. Encouraging saltmarsh growth at the expense of mudflat habitat may, in some locations, have considerable implications and require strong justification. Nonetheless, at locations where the mudflat is eroding, the need to stabilise the mudflat is becoming more pressing. For example, at Rumney Great Wharf, as further erosion of the Rumney, Awre and Northwick formations occurs, more of the middle Wentlooge peat deposits are exposed and subject to erosion (Armstrong et al., 2021). These peats likely contain a relatively high organic carbon stock (e.g., Chmura et al., 2003) and their erosion could cause the release of carbon sequestered within the peats up to 6,000 years ago.

Creating saltmarsh on mudflats will not only help stabilise eroding mudflats but could help to protect the pre-existing marsh and, in doing so, protect the organic carbon already stored within it (Smeaton et al., 2024), providing further justification for future site implementation. The construction of sedimentation fields may also help meet the saltmarsh extent increases targeted by coastal managers. For example, in England, the Environment Agency's Restoring Meadow, Marsh and Reef (ReMeMaRe, https://ecsa.international/rememare/restoring-meadow-marsh-and-reef-rememare) initiative is targeting a 15% increase in saltmarsh by 2043. Likewise, since the early 1990s, saltmarsh restoration in the United Kingdom through tidal inundation and depositing dredged sediment has resulted in the delivery of nearly 3,000 ha of new intertidal habitat (Ladd, 2021). However, this does not deliver enough habitat to offset the reduction in saltmarsh extent that is known to have occurred historically (Phelan et al., 2011), which must first be addressed before a net gain can be achieved.

## Conclusion

Sedimentation fields may mitigate some of the issues observed in other saltmarsh restoration or creation approaches (e.g., poor drainage, reduced plant abundance and diversity). However, we argue that they are not without human influence and remain shaped by both historic and ongoing human activity. The Rumney Great Wharf example demonstrates that such schemes should not be considered a "return to nature" but a managed evolution of coastal landscapes. There remains an expectation that restoration efforts provide a particular ecosystem service or deliverable, and it is still important to ensure that sites deliver additional benefits and that a functioning ecosystem is restored. It is also necessary to

protect the pre-existing saltmarsh habitat and reduce erosion rates. Further understanding of the physical functioning of sedimentation fields is clearly needed. This includes empirical data on the supply and delivery of sediment, and the influence of these factors on the establishment of saltmarsh habitat. The need for these datasets will become more pressing as coastal managers and decision-makers look to develop more sustainable approaches to coastal flood defence, coupled with the increased pressures of sea level rise and increased storm magnitude and frequency. This is particularly the case in locations where approaches such as managed realignment are not possible, and there is a need to increase wave attenuation to reduce pressures on flood defences and/or compensate for saltmarsh habitat loss.

**Open peer review.** To view the open peer review materials for this article, please visit http://doi.org/10.1017/cft.2025.10020.

**Data availability statement.** Data availability is not applicable to this article as no new data were created or analysed in this study.

**Acknowledgements.** The authors would like to thank two anonymous reviewers for their supportive and insightful comments, which have helped us to improve the manuscript. We also thank Martin Bell (University of Reading) for a useful discussion of the concept for this article.

**Author contribution.** Both authors contributed to the development, writing and organisation of the article.

**Financial support.** This research received no specific grant from any funding agency, commercial or not-for-profit sectors.

**Competing interests.** The authors declare none.

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
