## [Reviewer Report]

Comments to the author

Overall, I think this is a key and important piece of work that the community urgently needs. As we continue to debate the pros and cons of NbS, we often pay little attention to the history and heritage of sites – a point brought forward excellently in this work. I commend the authors on the clarity of the text. Sentences are concise, clear and terms are explained very well. This paper has broad interest to both the scientific community and policy makers.

Some more recent global literature on the state of the art should be included in the introduction. A short visual timeline of changes in the area would also strengthen the paper. I also provide some other minor points for improvement below.

Impact statement

The impact statement could be tailored a little bit more to policymakers (as this is a good intended audience for such an interesting paper). Good to compare with managed realignment but I think it would be key to state here that it does not require an investment in conversion of land use type (which is generally socially or culturally unacceptable). The term ‘the required ecosystem functioning’ is also potentially a bit vague – perhaps a statement on the saltmarsh quality (e.g. it is diverse in species and can provide habitats for all normal salt marsh fauna).

Abstract

Experiencing losses globally point here to a key cause (urbanization or embankment mostly).

Add a one sentence description of sedimentation fields at L37.

L47 – it feels strange to shift tense here after the previous sentence in the past tense.

Introduction

Generally this section is clear and reads well. It would be useful to make a clear distinction here on what you define as a sedimentation field. This can be interpreted in three key ways: 1) an area where sedimentation happens, 2) an area where you force sedimentation to happen using e.g. structures – make it clear whether structures are used for sediment trapping or not or 3) an area where sediment is laid out to settle and dry (e.g. the Ems Dollard sedimentation fields). Hence, it is important to make your definition very clear up front. L84-85 hints at this but needs more detail.

L87-88 – consider rephrasing this is not quite clear. It does not necessarily follow logically that you are dealing with only created sedimentation fields on sites which are now no longer salt marsh.

L89-90 – this is a key new hypothesis of this paper – very nice!

L91 – in general ‘we evidence our argument’ reads a bit GenAI-ish – I would replace it with we argue this using the case of…

Additional literature to include:

It would be useful to include some of the recent Campbell paper (https://www.nature.com/articles/s41586-022-05355-z#:~:text=Globally%2C%20an%20area%20of%20salt,2%2C172.07)%20km2%20(Fig.) in your introduction. This can highlight the global state of the art and reemphasize the scale of the issue.

You can also consider including some other literature on the sedimentation problem including: Cox et al. 2022 which looks at sedimentation structures https://www.sciencedirect.com/science/article/pii/S0921818122000637

Also interesting to consider the tidal replicate gates as an alternative method (https://www.nature.com/articles/s41598-021-80977-3)

And include Wolters reflection on land use types to strengthen that argument https://besjournals.onlinelibrary.wiley.com/doi/full/10.1111/j.1365-2664.2008.01453.x

Human activity

Very nice photographs, text is clear. A graphical timeline of changes would be really useful.

L173 – do you know if this was native spartina? In China there is the case of spartina planting in the 1950s which was invasive and there have been recent drastic effects (see https://www.sciencedirect.com/science/article/pii/S0034425725002172?ssrnid=4982154&dgcid=SSRN_redirect_SD)

L178 – extent extended

L203 – something goes wrong with the reference here.

L206-206 – this is a key statement of your hypothesis and needs extending. Where and why is managed realignment not suitable everywhere? And is it ever likely that a choice must be made between the two options? If they essentially operate in different locations – then the comparison is not needed.

L211-225 – I understand what you aim for here, but give it a bit more context in terms of the broader field e.g. the rewilding discussion.

Future sedimentation fields

L230 – be clear here and everywhere what you mean by restoration. Is it land raising/stopping erosion, is it restoration of former marsh extent (then what do you take as ‘former’) or some form of ecological restoration goal e.g. species diversity

L235 – this is quite a bold statement. I agree in a sense that you are ‘helping nature do its job’ – and it would not happen otherwise, but you are capturing an ongoing natural process. Maybe consider rephrasing in a more subtle way. Or argue that because you need to keep doing maintenance, it can never self sustain and be truly natural.

L237 – if you open this door of ecosystem services, better go through it fully and discuss species diversity, resilience for wave action etc.

L244 – see Cox et al. 2022 and references therein of the Indonesian case with field sites and monitoring. Also ongoing work of Iris Moller (https://www.sciencedirect.com/science/article/pii/S0272771423002421). Maybe also that of Andreas d’Alpaos and his group. Either remove this statement or place in the context of such studies.

L274 – after this very nice section on ecosystems – it would be good to see something on social and cultural acceptance of such measures. These are typically well accepted measures. But one of their downfalls is their poor maintenance (money runs out, governance structures lack accountability etc.) – add something about this to bring some societal relevance to the forefront and beyond the case level.

---

## [Reviewer Report]

General comments:

I found this manuscript to be a very interesting and clearly written perspective. The authors challenge a common assumption in restoration ecology by arguing that sedimentation fields are not really a return to “natural” conditions but another way that people shape coastal ecosystems. I think it makes a meaningful contribution to the debate in restoration ecology whether to use historic ecosystem states and if so, which one.

I also think the paper’s outcome is strongly tied to semantics, but that doesn’t diminish its importance. Personally, I prefer using the term ‘(re)creation’ rather than ‘restoration’, to avoid discussion about the ecosystem state being targeted. In most situations a return to natural and thus pristine state has become impossible. A brief discussion whether “(re)creation” would be a more accurate description of sedimentation fields, and more broadly how terminology shapes management and policy perceptions, could sharpen the conceptual contribution of the paper.

The use of Rumney Great Wharf as a case study strengthens this discussion, situating sedimentation field implementation within a broader and detailed historical and geomorphological context, and made the argument more concrete. Adding additional case studies could further strengthen the discussion, although I recognize these are difficult to find, as sedimentation fields have not received much attention in general.

The manuscript is particularly relevant in light of the increasing policy emphasis on “nature-based solutions” and coastal adaptation triggered by climate change. Clear links to “Hold the Line,” habitat compensation, and carbon markets ensure the paper speaks directly to current management debates.

With minor adjustments, I believe the paper will be well suited for publication.

Specific comments:

Lines 55-69:

Only managed or unmanaged realignment are mentioned as ways to restore tidal inundation. However, there are more ways to (re)introduce the tide: Regulated Tidal Exchange systems that have been applied in the UK, Belgium, USA and probably more countries. Addition of the RTE technique and associated references would make the list more complete.

Lines 67-69:

“The manuscript lists several techniques for restoring intertidal habitat. However, since this paper focuses on sedimentation fields, I would expect references to studies where sedimentation fields have actually been applied or examined. This would help demonstrate how frequently and widely they have been used, and why they are relevant for tidal marsh restoration worldwide. Only in line 96 some other places (Wadden Sea) are mentioned (but not specifically). Indeed there are quite some sedimentation fields in the Waddensea, but also North Sea (Weser; Michaelis et al. 2024). Are there more? Providing more context on where sedimentation fields occur could situate the technique in a clearer international perspective.”

Lines 71-80:

This paragraph focuses on the deviations found after restoration efforts when compared to reference sites. It risks giving a skewed impression by only highlighting failures. However, I believe there have also been success stories following tidal restoration, which are not mentioned here. As written, the text gives the impression that such successes do not exist at all.

Lines 71-80:

Reduced hydrological connectivity seems to be a problem in more restoration sites (both MR and RTE) indeed (also Spencer et al. 2017 and Van Putte et al., 2020), but I don’t read about the effects on the biogeochemical cycle. In both abstract and conclusion anoxia is mentioned, but this is not described in the rest of the manuscript. Concerning biogeochemical effects the paper of Van Putte et al. 2025 (https://doi.org/10.1016/j.scitotenv.2024.178001) may also interest you.

Lines 176-186 – Figure 1:

It would be helpful when the figure included dates (and possibly color coding) to make the chronology clearer in space. For example: the first sedimentation field of 1989 could have the year shown inside, or be highlighted with a different color. In addition, the tems ‘previous fencing’ and ‘new fencing’ are a bit confusing as they stand. Adding the relevant years alongside these labels would make the figure easier to interpret.

Lines 317-318:

‘In some cases’ points out there are other cases. However, these were not mentioned in this paper.

Textual comments:

The writing is clear and well-structured. I found some little typos in the text:

Line 140:

‘Provide’ should be ‘provides’.

Please check the reference list.

The DOI of the references of Chmure et al, 2003, Dale & Arnall 2024, Ladd 2021, Smeaton et al. 2024, Spencer et al. 2017, Williams & Dale 2023 are not in the correct format (https://doi.org/https://doi.org/.....).

I think some sentences are a little long and could be broken up for readability/more clarity, but this is more about style than typos. The same applies to the placement of the words “however” and “therefore” in a sentence; while stylistic, small adjustments would make text easier to read.

Lines 155-158:

Original

“Behind the modern seawall, the reen and ditch habitat of the Gwent Levels, formed because of intertidal reclamation, is designated as a Site of Special Scientific Interest, supporting rare and endangered insect species such as Odontomyia ornata and Hydaticus transversalis.”

Suggestions

“Behind the modern seawall lies the reen and ditch habitat of the Gwent Levels, which was created through intertidal reclamation. This area is designated as a Site of Special Scientific Interest and supports rare and endangered insect species such as Odontomyia ornata and Hydaticus transversalis.”

or

“Behind the modern seawall, the Gwent Levels’ reen and ditch habitat—formed through intertidal reclamation—is designated as a Site of Special Scientific Interest and supports rare and endangered insects, including Odontomyia ornata and Hydaticus transversalis.”

Lines 314-320:

This paragraph can be written more to the point:

Suggestion:

Sedimentation fields may mitigate some issues observed in other restoration approaches (e.g., poor drainage, reduced plant abundance and biodiversity). However, in some cases they remain shaped by both historic and ongoing human activity. The Rumney Great Wharf example demonstrates that such schemes should not be considered a “return to nature” but a managed evolution of coastal landscapes.

Lines 322-326:

Very long and dense sentence.

I would suggest splitting this sentence into two parts or streamline it.

Streamlined suggestion:

“Although human activity may confound saltmarsh restoration through sedimentation fields, restoration projects are still expected to deliver ecosystem services and additional benefits while ensuring the recovery of a functioning ecosystem.”

I noticed that the words “however” and “therefore” are often placed in the middle of a sentence, which make the text read a little heavily. Below some examples, but there are more in whole manuscript.

Lines 211-213:

Original

Past human activity is still, however, likely to have influenced site formation processes, and hence the modern-day landscape, and should be accounted for when setting restoration targets.

Suggestion

“However, past human activity is still likely to have influenced site formation processes and, hence, the modern-day landscape, and should be accounted for when setting restoration targets.”

Lines 259-261:

Original

“It is, therefore, crucial that sedimentation fields are monitored to evaluate ecosystem functioning, including empirical assessments of the biogeomorphic processes.”

Suggestions

“Therefore, it is crucial that sedimentation fields are monitored to evaluate ecosystem functioning, including empirical assessments of biogeomorphic processes.”

or

“Sedimentation fields must therefore be monitored to evaluate ecosystem functioning, including empirical assessments of biogeomorphic processes.”

---

## [Editor Report]

This paper is well written and addresses an important topic. The reviewers provide good comments and suggested edits. I only have one main comment - 

My editorial comments relate to making the manuscript understandable to a wide global audience. For example I suggest for the Impact Statement and Abstract to include information on what is a “sediment field” as this is a regional approach/name that may not be familiar to our international journal audience. Also using terms like “managed realignment” is important to the manuscript, but please provide a definition in the Introduction. Globally that term can mean very different things. Please consider adding to the manuscript other examples to support your work, for example in the US features like this are called “silt fences” and are used to prevent erosion and the construction of sediment islands to trap sediment for accretion.

---

## [Reviewer Report]

Dear authors,

I believe you have addressed my comments thoroughly and clearly. All revisions made have improved the clarity and quality of the manuscript, and all concerns have been appropriately resolved. I am satisfied with the changes and and I am pleased to recommend acceptance.